# Assessing the impact of rising child poverty on the unprecedented rise in infant mortality in England, 2000–2017: time trend analysis

David Taylor-Robinson,[1,2] Eric T C Lai,[1] Sophie Wickham,[1] Tanith Rose,[1] Paul Norman,[3] Clare Bambra,[4] Margaret Whitehead,[1] Ben Barr[1]

[1]Department of Public Health and Policy, University of Liverpool, Liverpool, UK
[2]Section of Epidemiology, Department of Public Health, University of Copenhagen, Copenhagen, Denmark
[3]School of Geography, University of Leeds, Leeds, UK
[4]Institute of Health and Society, Newcastle University, Newcastle, UK

**Correspondence to**
David Taylor-Robinson;
dctr@liv.ac.uk

## ABSTRACT

**Objective** To determine whether there were inequalities in the sustained rise in infant mortality in England in recent years and the contribution of rising child poverty to these trends.

**Design** This is an analysis of trends in infant mortality in local authorities grouped into five categories (quintiles) based on their level of income deprivation. Fixed-effects regression models were used to quantify the association between regional changes in child poverty and regional changes in infant mortality.

**Setting** 324 English local authorities in 9 English government office regions.

**Participants** Live-born children under 1 year of age.

**Main outcome measure** Infant mortality rate, defined as the number of deaths in children under 1 year of age per 100 000 live births in the same year.

**Results** The sustained and unprecedented rise in infant mortality in England from 2014 to 2017 was not experienced evenly across the population. In the most deprived local authorities, the previously declining trend in infant mortality reversed and mortality rose, leading to an additional 24 infant deaths per 100 000 live births per year (95% CI 6 to 42), relative to the previous trend. There was no significant change from the pre-existing trend in the most affluent local authorities. As a result, inequalities in infant mortality increased, with the gap between the most and the least deprived local authority areas widening by 52 deaths per 100 000 births (95% CI 36 to 68). Overall from 2014 to 2017, there were a total of 572 excess infant deaths (95% CI 200 to 944) compared with what would have been expected based on historical trends. We estimated that each 1% increase in child poverty was significantly associated with an extra 5.8 infant deaths per 100 000 live births (95% CI 2.4 to 9.2). The findings suggest that about a third of the increases in infant mortality between 2014 and 2017 can be attributed to rising child poverty (172 deaths, 95% CI 74 to 266).

**Conclusion** This study provides evidence that the unprecedented rise in infant mortality disproportionately affected the poorest areas of the country, leaving the more affluent areas unaffected. Our analysis also linked the recent increase in infant mortality in England with rising child poverty, suggesting that about a third of the increase in infant mortality from 2014 to 2017 may be attributed to rising child poverty.

### Strengths and limitations of this study

► Use of national-level data shows that the unprecedented rise in infant mortality in England has disproportionately affected the poorest areas of the country, leaving the more affluent areas unaffected.

► Using linked area-level data we show that the recent increase in levels of child poverty was associated with about a third of the extra infant deaths in England in the period 2014–2017.

► Limitations include the lack of individual-level data and information on potentially mediating pathways, such as changes in health and social care spending on children, which may have contributed.

## INTRODUCTION

Infant mortality rate (IMR) has risen for the last 4 years in England, yet the role of increasing levels of child poverty in explaining these trends is unclear. Along with others,[1] we raised concern about recent rises in infant mortality in England in two letters to the *BMJ*.[2 3] We noted that this rise had occurred particularly among more disadvantaged children from routine and manual socioeconomic groups. Rising infant mortality is unusual in high-income countries, and international data show that infant mortality has continued to decline in most rich countries in recent years.[4] Infant mortality is a sensitive indicator of the changing overall health of societies, and as such acts as an early warning system for future adverse trends. There is therefore an urgent need to understand this extremely concerning trend in England.

Adverse trends in mortality have occurred across all age groups in England in recent years.[5] Several commentators have suggested that these could be due to austerity policies introduced in recent years, including cuts to National Health Service (NHS), local authority and public health services, and

changes to welfare benefits.[6][7] Since 2010, there have been sustained reductions in the welfare benefits available to families with children, including the abolition of child benefit and child tax credit for the third child or more; reductions in the value of tax credits and below-inflation up-rating of most working-age benefits; housing benefit reforms including the under occupancy charge (most commonly referred to as 'bedroom tax') and introduction of universal credit; and household caps on total benefit receipt (regardless of how many children are in the household).[8] These welfare changes have disproportionately affected the most deprived local authorities and regions[8] and have led to a rise in child poverty.[9] The impacts of these changes on trends in child health have not been considered. For example, a recent review of mortality increases by Public Health England did not consider the potential causes of increases in infant mortality.[5]

There is strong evidence that increased child poverty leads to deteriorating child health and increased infant deaths.[10–14] While relative child poverty declined between 2007 and 2013, we are now seeing increases: child poverty (defined as living in a household with income below 60% of the median household after housing costs) rose by two percentage points between 2014 and 2017, and it is projected to increase further through to 2022.[9] By 2017 there were 4.1 million children in England living in relative poverty, amounting to 30% of all English children. This compares with less than 10% of children in European countries such as Austria, Denmark, Finland, Iceland, Norway, Slovenia, Sweden and Switzerland.[15] The recent rises in infant mortality in England have occurred concurrently with these increases in child poverty (figure 1).

In this paper, we investigate whether there were inequalities in the sustained rise in infant mortality in England in recent years and the contribution of rising child poverty to these trends. First, we investigate whether infant mortality increased more in those parts of the country with the highest numbers of people receiving low income-related welfare benefits, as these are the areas that are most likely to have been adversely affected by recent changes in welfare policy. We estimate whether there was a significant change in the trend in infant mortality in those areas and the timing of this change. Second, we exploit regional differences in trends in infant mortality and child poverty to estimate the extent to which those regions with greater increases in child poverty experienced greater increases in infant mortality. We use these estimates to calculate the proportion of recent infant mortality increases that are potentially explained by increases in child poverty.

## METHODS

Annual Vital Statistics data for the number of registered infant deaths (<1 year of age) and live births for 324 lower tier local authorities (324 local areas) between 2000 and 2017 were obtained from the Office for National Statistics. We chose this period due to data availability and since it captures a contemporary period during which both infant mortality and child poverty rates have changed dramatically. Local authorities were grouped into five categories (quintiles) based on the income deprivation score of the 2015 Indices of Multiple Deprivation. This score is a non-overlapping count of the number of people in each local authority on low income and in receipt of means-tested benefits as a proportion of the population.[16]

Regional relative child poverty was measured as the proportion of children living in households in each region with less than 60% of contemporary national median household income, using the Households Below Average Income data provided by the Department for Work and Pensions.[17] This is the most widely used measure of poverty within the European Union and recognises that the experience of poverty is relative to standards of living that are considered normal within a society and that these average standards change over time.[14] As this measure is not available at the local authority level, we used data for the nine government office regions of England.[18] See online supplementary web appendix 1 for a summary of the data set.

### Patient and public involvement

Our study was informed by discussions with children and young people's (CYP) reference groups in Liverpool who encouraged us to undertake analyses focused on developing a better understanding of social factors that shape CYP's lives, such as child poverty. The results of our work are feeding into ongoing discussion with CYP about health inequalities, informing the UNICEF Child Friendly City programme in Liverpool.

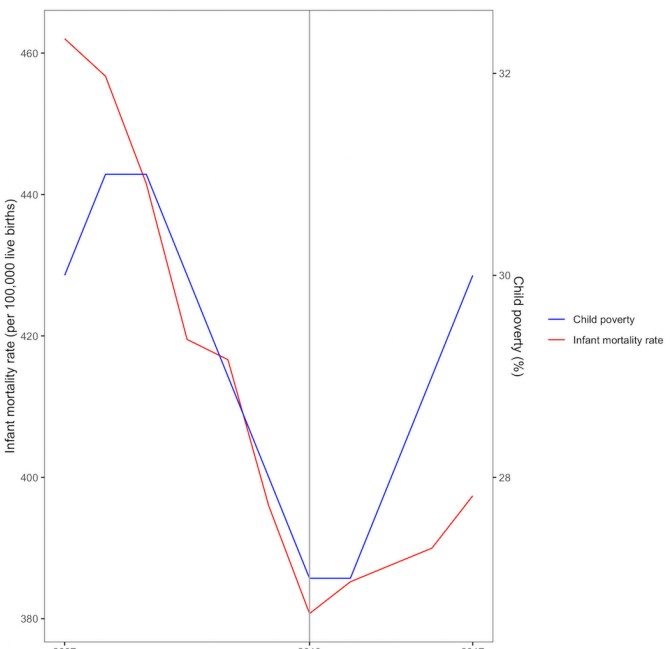

**Figure 1** Trends in infant mortality and child poverty in England, 2007–2017.

## Statistical analysis

The statistical analysis proceeded in three steps. First, we assessed descriptive trends in infant mortality between 2000 and 2017 for the five groups of local authorities categorised by their level of income deprivation (see above). Second, we tested whether there was a statistically significant change in the infant mortality trend during this time, the timing of that change in trend and whether any change in trend differed by the level of income deprivation. To do this we used longitudinal local authority area-level data to estimate a segmented mixed-effects regression model for infant mortality, including two linear spline terms for time interacted with income deprivation quintile, with a breakpoint indicating the change in trend. We used an iterative search procedure to identify which breakpoint provided the best fit for the data,[19 20] in other words identifying at which time point a change in trend occurred. This model included random effects (intercepts) at the local authority level to take account of the longitudinal nature of the data (see online supplementary web appendix for details). Second, we used this model to estimate whether there had been a significant change in trend in recent years, whether this had led to widening inequalities between local authority areas based on their level of income deprivation and to estimate the level of excess mortality that was attributable to any change in trend. Excess mortality was estimated as the marginal difference between the observed trend and that which would be predicted from the model if pre-existing trends had continued, using the *margins* commands in Stata V.14.[21]

In the third step of the analysis, we assessed the association of changes in regional child poverty and regional infant mortality. As there is potential confounding from unobserved factors that vary between regions, or national trends in unobserved factors that affect all regions, we used a fixed-effects approach to remove between-region differences and national trends.[22 23] This conservative approach involves including dummy variables for each region and year to assess the association between deviations from the average rate of change in poverty and deviations from the average rate of change in infant mortality in each region (see online supplementary web appendices 2–3 for further details). This method means that the estimated association between child poverty and infant mortality cannot be confounded by any time-invariant differences between regions or any national trend that affects all regions, providing an estimate that is likely to reflect a causal association. We aligned annual regional child poverty rates for financial years to infant mortality registered in the calendar year in which the financial year ended. In other words, 2016–2017 child poverty data were aligned with 2017 infant mortality data, meaning that child poverty measures were lagged by 9 months compared with infant mortality measures. We used robust clustered SEs to ensure that they are robust to serial correlation in the data. This model was used to estimate the number of child deaths attributable to child poverty

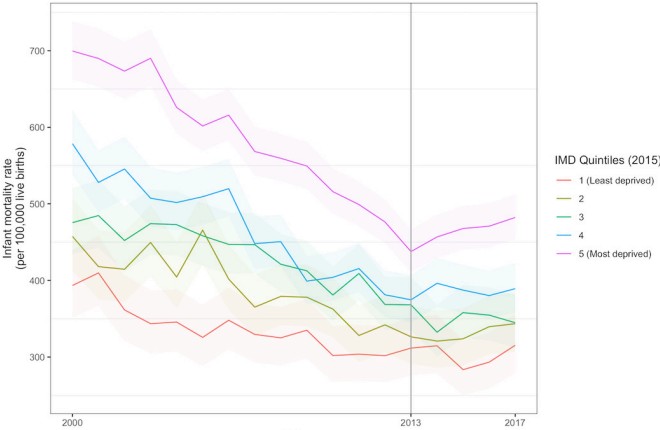

**Figure 2** Infant mortality trend by deprivation quintile of local authority district, 2000–2017, with 95% binomial CIs. IMD, Index of Multiple Deprivation.

increases in recent years, by estimating the marginal difference between the observed trend and that which would be predicted from the model if child poverty had not increased.[21] All models were estimated using Stata V.14 and R V.3.5.1. As a robustness test we repeated our analysis stratified by neonatal and postneonatal deaths.

## RESULTS

Infant mortality had been falling for all groups of local authority areas since 2000, with greater decreases in the most income-deprived areas, reducing inequalities. From 2013, this trend changed, and infant mortality increased particularly in the most income-deprived local authority areas (figure 2). In online supplementary web appendix 4 we have provided a link to a visualisation of the geographical distribution of the IMR data for England.

The breakpoint analysis also showed a significant change in trend from 2013 (online supplementary web appendix). Table 1 shows the results of the regression model for trends in infant mortality since that time. From 2013 to 2017 there was a dramatic reversal of the pre-existing trend, with an additional annual increase of 24 deaths per 100 000 live births (95% CI 6 to 42) in the most deprived quintile 5 compared with the 2000–2013 trend. A significant change in trend was also observed for quintile 4, while there was no significant change from the previously declining trend in the most affluent quintile. The gap in infant mortality between the most income-deprived and most affluent quintile increased by 52 deaths per 100 000 births (95% CI 36 to 68). Based on these trends we estimated that there were an additional 570 excess deaths (95% CI 200 to 944) in the period 2014–2017 than would have been expected if the historical trends had continued.

In the fixed-effects analysis each percentage point change in child poverty was associated with a change in infant mortality of 5.8 deaths per 100 000 live births (95% CI 2.4 to 8.9). From this model we estimate that 172 deaths (95% CI 74 to 266) between 2014 and 2017 were

**Table 1** Segmented regression model results for trends in infant mortality at lower tier local authority level in 2014–2017 (per 100 000 live births per year)

| Level of income deprivation | Annual change (deaths per 100 000) in infant mortality (2014–2017) relative to previous trend (2000–2013) (95% CI) | P value for change in trend from previous period |
|---|---|---|
| Quintile 1 (most affluent) | −4.19 (−16.91 to 8.53) | 0.518 |
| Quintile 2 | 15.65 (−2.27 to 33.57) | 0.087 |
| Quintile 3 | 3.89 (−14.03 to 21.81) | 0.67 |
| Quintile 4 | 20.5 (2.58 to 38.42) | 0.025 |
| Quintile 5 (most deprived) | 24.14 (6.22 to 42.05) | 0.008 |

Estimates based on random-effects regression model using local authority panel data set of infant mortality from 2000 to 2017, n=5832 local authority years. See online supplementary web appendices 5–9 for full model output and checks.

attributable to increases in child poverty, almost a third of the overall rise in infant mortality over that period. Repeating our analysis for neonatal and postneonatal deaths showed similar results, with a change in trend impacting the most disadvantaged quintiles (see online supplementary web appendices 10 and 11).

## DISCUSSION

Since 2013, infant mortality has increased in England and there have been an additional 570 infant deaths over 4 years (2014–2017) compared with what would have been expected based on recent historical trends. These excess deaths have largely occurred in the most disadvantaged areas, increasing inequalities. Our study estimated that the recent increase in levels of child poverty was associated with 172 (about a third) of the extra infant deaths in England in the period 2014–2017.

### Limitations of the study

Before evaluating the implications of our findings, we note several important limitations. First, due to the absence of individual-level data, we undertook an observational analysis at the population level. Lack of data on levels of child poverty at a small area level meant that we undertook our analysis of the effects of child poverty using child poverty estimates for nine regions within England. This meant we could take advantage of regional differences in trends over time and use a fixed-effects approach to adjust for time-invariant confounding variables. We were not able, however, to estimate the effect of increases in child poverty on trends in infant mortality for groups of local authorities within these regions. As our main analysis was based on aggregate data, however, we cannot identify whether the additional infant deaths were

in the same families who have experienced a rise in child poverty. It is also possible that the association between child poverty and infant mortality in our analysis was due to trends in unobserved time-varying confounding factors that varied between regions.

Second, we measured child poverty using the relative poverty measure used by the Department for Work and Pensions in England because it is used internationally as well as in the UK. Furthermore, whether or not relative measures of income poverty effectively reflect children's life chances has been the focus of policy debates in the UK.[24] This measure, however, is a simplification of underlying income trajectories whereby families with children move in and out of poverty over time. Furthermore, the measure we have used is likely to underestimate the extent of the experience of poverty in England. A recent report published by the Social Metrics Commission outlining a new measure of poverty which extends beyond income-based poverty suggests that there are 4.5 million children living (32.6%) in poverty in the UK.[25] In addition, the Joseph Rowntree Foundation's has shown that the proportion of children living below the minimum income standard has increased from 39% to 45% (an increase of about 1 million) between 2008–2009 and 2014–2015.[26] Further studies could usefully explore the relationship between a continuous measure of income and aspects of child health using individual-level data in order to identify any policy-relevant threshold effects.

Third, we did not have data on cause of death and were not able to investigate the factors mediating the association between rising child poverty and rising infant mortality, such as birth characteristics (eg, preterm birth, maternal age and maternal smoking) and postnatal factors (eg, breast feeding and childcare in the first year of life).[27 28] Fourth, it is important that further research investigates the potential role of parental characteristics, gestational age and other risk factors for child mortality in explaining the observed trends, in addition to assessing how changes in health and social care spending on children may have contributed.[6 7]

### Comparison with previous studies

To our knowledge this is the first study to explore both the inequalities in the unprecedented rise in infant mortality in England on the basis of area deprivation and the reasons behind this rise. While the UK government recently commissioned a report into adverse mortality trends, infant mortality was not considered in any detail.[5] Our analysis corroborates our previous analysis of trends in infant mortality on the basis of parental occupational social class, which suggested that IMR was rising particularly in the most disadvantaged social groups.[2 3] A recent report by the Royal College of Paediatrics and Child Health (RCPCH) also noted the reversal of over 100 years of declining infant mortality in England and Wales, and that rises in infant mortality were most striking in the deprived portion of the population.[27] The RCPCH further assesses various scenarios of infant mortality, for

instance showing that even if infant mortality begins to decline again at its previous rate, rates will be 80% higher than the EU15+ average in 2030. Our study uses the most up-to-date infant mortality data and suggests that, rather than just stalling, infant mortality is rising sharply in the most disadvantaged areas.

In addition to increasing inequalities in infant mortality, social inequalities in numerous aspects of health in England appear to be rising.[7 19 29 30] Barr and colleagues[19] demonstrated a reversal in trends in inequalities in life expectancy. Two analyses from the Global Burden of Disease collaboration have shown both a stagnation in the improvement in life expectancy in England and increasing inequalities since 2010.[29 30] Furthermore Bennett et al[29] identify deaths in children younger than 5 years as a major contributor to inequality in life expectancy. A further analysis has demonstrated an increase in the North–South divide in health in England.[7]

Our study shows that rising income poverty may be contributing to rising infant mortality. This finding is likely to be generalisable to other high-income settings, since numerous studies conducted in Western Europe and the USA have shown an association between social disadvantage and infant mortality.[31–34] Furthermore, numerous studies have suggested that there is a causal link between increasing child poverty and a deterioration in various aspects of child health and development.[10–14] For example, a recent systematic review by Cooper and Stewart[13 35] concluded that 'poorer children have worse cognitive, social-behavioural and health outcomes in part because they are poorer, and not just because poverty is correlated with other household and parental characteristics'. Their review demonstrated clear causal links between income poverty and a range of child health outcomes, including only studies that applied quasi-experimental designs. Komro and colleagues,[36] for example, used a difference-in-differences research design to assess the impact of state-level minimum wage in the USA on infant mortality. The authors show that each dollar increase in the minimum wage above the federal level was associated with a 1%–2% decrease in low birthweight births and a 4% decrease in postneonatal mortality. Additionally, the results of a longitudinal analysis of social welfare expenditure data from 19 Organisation for Economic Co-operation and Development countries over the time period from 1980 to 2010 showed that cash benefits for families have positive effects on reducing infant mortality, with a $10 increase in family cash allowances per child predicting a reduction in infant mortality by 4% (p=0.007).[12] Previous research has shown that expansions to social security nets and public services have decreased inequalities in life expectancy,[19] IMRs[37] and mortality amenable to healthcare.[38] Robinson et al,[37] for example, found that the English health inequalities strategy (2000–2010) was associated with decreases in inequalities in infant mortality in England. Similarly, Krieger et al[39] found that the 1960s' 'War on Poverty' led to decreases in inequalities in IMR in the USA.

## Policy implications

Our analysis has important implications in the context of the projected ongoing increase in levels of child poverty in the UK. For the first time in nearly 40 years, there has been a sustained increase in infant mortality in the poorest areas. English regions with the largest rises in child poverty have had the largest increases in infant mortality. Our analysis suggests that the weakened social protection safety net—leading to rising levels of child poverty—may be contributing to this rise in mortality in the most disadvantaged infants. Our analysis suggests that the rise in child poverty explains approximately a third of this rise. This could be an underestimate of the impact if the measures of child poverty used are an underestimate of the true increase in disadvantage facing children (see above). It may also be the case that other policy changes occurring at this time, such as the real terms cuts to NHS, local authority children's services, social care and public health budgets, are also contributing to the rise in mortality as has been suggested elsewhere.[6 7]

Infant mortality is an uncommon event and represents the most severe tip of the iceberg in terms of the impact of social conditions on health. It can, however, act as an early warning system or litmus test of the overall health of societies. Rising poverty is likely to be having a myriad of adverse impacts on other aspects of child health that will have repercussions across the life course.[35] In the context of increasing health inequalities in England, policies that reduce poverty and social inequalities and investing in child health and social care are likely to reduce the occurrence of infant mortality and that of many other adverse child health outcomes.[29]

This rise in mortality in the most disadvantaged children is unprecedented and requires urgent action by national and local governments and the health and social care system. It is likely that the rise in child poverty is an important factor contributing to this trend. As the United Nations has recently highlighted, rising poverty in the UK is a political choice,[40] and it is time for the government to reverse this trend, establishing a welfare system that protects children from poverty.

**Contributorship statement** DT-R is the lead author and guarantor. DT-R and BB planned the study and led the drafting and revising of the manuscript. DT-R, BB and ETCL conducted the analysis. DT-R, BB, ETCL, SW, TR, PN, CB and MW contributed to interpretation of the data, drafting of the manuscript and revisions. All authors agreed on the submitted version of the manuscript.

**Funding** DT-R and ETCL are funded by the MRC on a Clinician Scientist Fellowship (MR/P008577/1). SW is supported by a Wellcome Trust Society and Ethics fellowship (grant number 200335/Z/15/Z). BB and TR are supported by the National Institute for Health Research (NIHR) Collaboration for Leadership in Applied Health Research and Care (CLAHRC NWC) and the NIHR School for Public Health Research. The MRC, Wellcome Trust and NIHR had no role in the study design, data collection and analysis, decision to publish, or preparation of the manuscript. This report is an independent research arising from research supported by the MRC, Wellcome Trust and NIHR. The views expressed in this publication are those of the authors and not necessarily those of the NHS, the NIHR, or the Department of Health and Social Care.

**Competing interests** None declared.

**Patient consent for publication** Not required.

**Provenance and peer review** Not commissioned; externally peer reviewed.

**Data availability statement** Data are available upon reasonable request.

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
