## [Reviewer comments · BMJ Open]

ARTICLE DETAILS

TITLE (PROVISIONAL)	Assessing the impact of rising child poverty on the unprecedented rise in infant mortality in England 2000-17: time trend analysis
AUTHORS	Taylor-Robinson, David; Lai, Eric; Wickham, S; Rose, Tanith; Norman, Paul; Bambra, Clare; Whitehead, Margaret; Barr, Ben

VERSION 1 – REVIEW

REVIEWER	Jenny Garcia Institut National d'études Démographiques INED France
REVIEW RETURNED	14-Feb-2019

GENERAL COMMENTS	The authors provide new evidence on the influence of poverty on the changes in infant mortality rate (IMR) in England. This is an important study because a) England has been showing detrimental health and mortality outcomes in recent years, b) there is scarce evidence about the relation between relative poverty and IMR trends, c) contribute to the ongoing debate about the impact of the weakening of the welfare state in health outcomes. The major strength of this study is the usage of the most up-to date data to estimate the effect in the reversing IMR trend attributable to increasing poverty. The evidence found is likely to be conservative due to the index chosen to measure poverty, as the authors acknowledge. While the results are compelling, it would be interesting to see them geographically located. Additionally, a comparison of the health system funding trends among the local authority districts would give a complementary idea of possible regional differences in the welfare changes. This is because socioeconomic variables predict only a small proportion of the variation in infant mortality, and environmental, community and health system characteristics might be also affecting IMR trends.
--

REVIEWER	Carla Blázquez-Fernández Universidad de Cantabria (Spain)
REVIEW RETURNED	25-Feb-2019

GENERAL COMMENTS	Contribution of interest but not accepted in its actual state. The article requires some modifications. Authors should reinforce several explanations: * Authors should justified period under analysis (why only 17 years?)* Why not considering cause of death?* Do authors work so with macro data and no with micro data? That is to say, authors are not linking mortality with the
---

	corresponding household and so with its income. It would be convenient to draw conclusions... * At this regard, factors determining the rise in infant mortality go beyond income inequality: i.e. What about country of born from parents? Other features could explain that figures observed... Authors say something in their limitations but it does not be enough. Considering all, policy implications remain empty. * Besides, authors should include future extensions and extrapolations to other areas or countries. How could results be extrapolated?
--	--

REVIEWER	Professor Colin Pritchard Bournemouth University , UK.
REVIEW RETURNED	12-Mar-2019

GENERAL COMMENTS	This is a very important paper, crucially, the minor limitations are well recognised and explained, and an ingenious use of time trend analysis. Rightly the tight focus is upon England but there is evidence that in terms of international comparison there is evidence that the UK has the third highest relative poverty and fifth highest children mortality of twenty-one Western countries (Children & Society, 29:138-144).The authors and BMJ Open are to be congratulated for highlighting the current situation, which is the socio-economic background in which services operate.
--

REVIEWER	Ayaz Hyder The Ohio State University, United States
REVIEW RETURNED	28-Mar-2019

GENERAL COMMENTS	This manuscript is an ecological study on the association between child poverty and infant mortality. The authors used longitudinal data on child poverty and IMR at the local authority level in this study. The authors found a relationship between declining and then rising levels of child poverty and IMR. Overall, this a a reasonable study but I have several hesitations regarding the claims of it as currently written.  1. Page 11, line 28. The claim by the authors that there is a causal association is not supported by their results. They also claim that previous studies show strong evidence of this relationship and my assessment of these previous studies does not bear this out. In public health, evidence-based interventions are highly encouraged yet the authors do not sufficiently provide such evidence upon which they base their policy recommendations. The use of an individual-level analysis in support of evidence derived from an ecological analysis would provide much stronger support for the purported association between child poverty and infant mortality. 2. The use of infant mortality as an indicator for overall health of societies makes sense but in England it may be more useful to look at specific aspects of IMR such as postneonatal and neonatal infant mortality. The set of risk factors associated with each of these types of IMR do not always overlap and I suggest the authors repeat the analysis to check for the sensitivity of their findings to type of IMR. 3. Are there any differences in women's characteristics, such as parity, age, and the baby's characteristics, such as birth weight, gestational age, that could further be used to stratify the trends
--

	over time. It is not very informative to combine these data into a single value of IMR for each year by local authority. 4. In the discussion, the authors do not explain why annual changes (Table 1) in quintile 2 are comparable to quintile 4. Yes, the p-values are not significant in one but not the other but why do we observe these similar magnitudes of change in quite opposite levels of income quintile. Also, the authors do not make a clear case for using child poverty as the predictor whereas other studies have used family income and other measures of social disadvantage. Why should policy makers address child poverty specifically over a number of other measures?
--	---

VERSION 1 – AUTHOR RESPONSE

Reviewer: 1

Reviewer Name

Jenny Garcia

Institution and Country

Institut National d'études Démographiques INED
France

Please state any competing interests or state 'None declared':

None competing interests

Please leave your comments for the authors below

The authors provide new evidence on the influence of poverty on the changes in infant mortality rate (IMR) in England. This is an important study because a) England has been showing detrimental health and mortality outcomes in recent years, b) there is scarce evidence about the relation between relative poverty and IMR trends, c) contribute to the ongoing debate about the impact of the weakening of the welfare state in health outcomes. The major strength of this study is the usage of the most up-to date data to estimate the effect in the reversing IMR trend attributable to increasing poverty. The evidence found is likely to be conservative due to the index chosen to measure poverty, as the authors acknowledge.

Response: Thanks

While the results are compelling, it would be interesting to see them geographically located.

Response: Thanks, we have added a map to the to show the distribution of infant mortality across England

"P8 In the appendix we have provided a link to a visualisation of the geographical distribution of the IMR data for England."

Appendix: Web appendix 10: Link showing show the distribution of infant mortality across England
<https://dblalex.carto.com/builder/c81de0d5-f893-4109-9471-b4b55afa351e/embed>

Additionally, a comparison of the health system funding trends among the local authority districts would give a complementary idea of possible regional differences in the welfare changes. This is

because socioeconomic variables predict only a small proportion of the variation in infant mortality, and environmental, community and health system characteristics might be also affecting IMR trends.

Response: Thanks, this is an important question, and as part of another project we are compiling area linked data on different aspects of spending on children, but we don't have this at the moment. We have added a sentence about this in the discussion. The main focus of this paper was to assess whether infant mortality has risen most in the poorest areas, and to relate any changes in IMR to changes in child poverty.

P11. "Fourthly, it is important that further research investigates the potential role of parental characteristics, gestational age and other risk factors for child mortality in explaining the observed trends, in addition to assessing how changes in health and social care spending on children may have contributed. [6,7]"

Reviewer: 2

Reviewer Name

Carla Blázquez-Fernández

Institution and Country

Universidad de Cantabria (Spain)

Please state any competing interests or state 'None declared':

None declared

Please leave your comments for the authors below

Contribution of interest but not accepted in its actual state. The article requires some modifications.

Authors should reinforce several explanations:

* Authors should justified period under analysis (why only 17 years?)

Response: Thanks – we have added a sentence about this – we only have access to the exposure and outcome over this period, a period over which both poverty and infant mortality have changed considerably, falling then rising.

P6 "We chose this period due to data availability and since it captures a contemporary period during which both infant mortality and child poverty rates have changed dramatically."

* Why not considering cause of death?

Response: Thanks – unfortunately we do not have the data on cause of death, we have noted this in the limitations

P10. "Thirdly, we did not have data on cause of death, and were not able to investigate the factors mediating the association between rising child poverty and rising infant mortality, such as birth characteristics (for example preterm birth, maternal age and maternal smoking) and post-natal factors (for example breastfeeding and child care in the first year of life)."

* Do authors work so with macro data and no with micro data? That is to say, authors are not linking mortality with the corresponding household and so with its income. It would be convenient to draw conclusions...

Response: Thanks. We do not have access to individual level micro-data. In the absence of this we have used area based measures of child poverty; linked to small area level linked data on child mortality. We note this in the limitations of the study.

P10. "Firstly, due to absence of individual level data, we undertook an observational analysis at the population level."

P10 "As our main analysis was based on aggregate data, however, we cannot identify whether the additional infant deaths were in the same families who have experienced a rise in child poverty."

* At this regard, factors determining the rise in infant mortality go beyond income inequality: i.e. What about country of born from parents? Other features could explain that figures observed... Authors say something in their limitations but it does not be enough. Considering all, policy implications remain empty.

Response: Thanks. We have added a sentence about this in the limitations and strengthened the policy implications section.

P11. "Fourthly, it is important that further research investigates the potential role of parental characteristics, gestational age and other risk factors for child mortality in explaining the observed trends, in addition to assessing how changes in health and social care spending on children may have contributed. [6,7]"

* Besides, authors should include future extensions and extrapolations to other areas or countries. How could results be extrapolated?

Thanks. We have added a sentence about this

P12. "Our study shows that rising income poverty may be contributing to rising infant mortality. This finding is likely to be generalisable to other high-income settings, since numerous studies conducted in Western Europe and the USA have shown an association between social disadvantage and infant mortality."

Reviewer: 3

Reviewer Name

Professor Colin Pritchard

Institution and Country

Bournemouth University , UK.

Please state any competing interests or state 'None declared':

None

Please leave your comments for the authors below

This is a very important paper, crucially, the minor limitations are well recognised and explained, and an ingenious use of time trend analysis.

Rightly the tight focus is upon England but there is evidence that in terms of international comparison there is evidence that the UK has the third highest relative poverty and fifth highest children mortality of twenty-one Western countries (Children & Society, 29:138-144). The authors and BMJ Open are to be congratulated for highlighting the current situation, which is the socio-economic background in which services operate.

Response: Thanks for the positive feedback

Reviewer: 4

Reviewer Name

Ayaz Hyder

Institution and Country

The Ohio State University, United States

Please state any competing interests or state 'None declared':

None declared

Please leave your comments for the authors below

This manuscript is an ecological study on the association between child poverty and infant mortality. The authors used longitudinal data on child poverty and IMR at the local authority level in this study. The authors found a relationship between declining and then rising levels of child poverty and IMR. Overall, this is a reasonable study but I have several hesitations regarding the claims of it as currently written.

1. Page 11, line 28. The claim by the authors that there is a causal association is not supported by their results. They also claim that previous studies show strong evidence of this relationship and my assessment of these previous studies does not bear this out. In public health, evidence-based interventions are highly encouraged yet the authors do not sufficiently provide such evidence upon which they base their policy recommendations. The use of an individual-level analysis in support of evidence derived from an ecological analysis would provide much stronger support for the purported association between child poverty and infant mortality.

Response: Thanks, we have removed this sentence and have simply noted that a recent systematic review has concluded that the evidence from quasi-experimental studies suggests that there is causal link between rising child poverty and adverse child health outcomes.

P12. "For example, a recent systematic review by Cooper and Stewart concluded that "poorer children have worse cognitive, social-behavioural and health outcomes in part because they are poorer, and not just because poverty is correlated with other household and parental characteristics."

2. The use of infant mortality as an indicator for overall health of societies makes sense but in England it may be more useful to look at specific aspects of IMR such as postneonatal and neonatal infant mortality. The set of risk factors associated with each of these types of IMR do not always overlap and I suggest the authors repeat the analysis to check for the sensitivity of their findings to type of IMR.

Response: Thanks – we have done this as a sensitivity analysis which is now in the appendix, and our conclusions are the same.

P9. “Repeating our analysis for neonatal and post-neonatal deaths showed similar results, with a change in trend in the most disadvantaged quintile for both outcomes, though the change was most pronounced for neonatal deaths (see appendix).”

3. Are there any differences in women's characteristics, such as parity, age, and the baby's characteristics, such as birth weight, gestational age, that could further be used to stratify the trends over time. It is not very informative to combine these data into a single value of IMR for each year by local authority.

Response: Unfortunately we do not have access to this demographic data at small area level for use in our analysis, but this will be the subject of further analyses that we plan to undertake in the future, for example using individual level micro-data available in Wales. We have undertaken the sensitivity analysis above, stratifying my neonatal and post-neonatal deaths.

4. In the discussion, the authors do not explain why annual changes (Table 1) in quintile 2 are comparable to quintile 4. Yes, the p-values are not significant in one but not the other but why do we observe these similar magnitudes of change in quite opposite levels of income quintile.

Response: Thanks. Our interpretation of the data is that there has not been any significant change in trend except for the two most disadvantaged quintiles (4 and 5). Plotting the estimates and their confidence intervals (see below) suggests that the data are compatible with a monotonic dose response relationship with deprivation. We have added this plot to the appendix materials.

Also, the authors do not make a clear case for using child poverty as the predictor whereas other studies have used family income and other measures of social disadvantage. Why should policy makers address child poverty specifically over a number of other measures?

Response: Thanks, we have added a sentence about this – we are particularly interested in child poverty since this is of current policy importance in the UK.

P10. Secondly, we measured child poverty using the relative poverty measure used by the Department for Work and Pensions in England, because it is used internationally, as well as in the UK. Furthermore, whether or not relative measures of income poverty effectively reflect children's life chances has been the focus of policy debates in the UK.[24]

We have added a sentence to suggest that studies should also explore the relationship with income across the distribution. This requires microdata, but can be done using data linkage, for example in Nordic countries.

P11. “Further studies could usefully explore the relationship between a continuous measure of income and aspects of child health using individual level data, in order to identify any policy relevant threshold effects.”

VERSION 2 – REVIEW

REVIEWER	Carla Blázquez Fernández Universidad de Cantabria (Spain)
REVIEW RETURNED	27-May-2019

GENERAL COMMENTS	The revised version has solved the problems I had pointed out in my first revision. I think the paper can be accepted for publication.
--

REVIEWER	Ayaz Hyder The Ohio State University, USA
REVIEW RETURNED	18-Jun-2019

GENERAL COMMENTS	Given the findings in Web Appendix 10, it would be helpful to plot the two types of IM in Web Appendix 9 so check whether the monotonic changes observed overall are really for all IM deaths or only applicable to neonatal IM.
--

VERSION 2 – AUTHOR RESPONSE

Reviewer: 2

Reviewer Name

Carla Blázquez Fernández

Institution and Country

Universidad de Cantabria (Spain)

Please state any competing interests or state 'None declared':
None declared

Please leave your comments for the authors below

The revised version has solved the problems I had pointed out in my first revision. I think the paper can be accepted for publication.

Reviewer: 4

Reviewer Name

Ayaz Hyder

Institution and Country

The Ohio State University, USA

Please state any competing interests or state 'None declared':
None declared

Please leave your comments for the authors below

Given the findings in Web Appendix 10, it would be helpful to plot the two types of IM in Web

Appendix 9 so check whether the monotonic changes observed overall are really for all IM deaths or only applicable to neonatal IM.

Response: Thanks – we have done this, and added a signposting note in the main document

P9: “Repeating our analysis for neonatal and post-neonatal deaths showed similar results, with a change in trend impacting the most disadvantaged quintiles (see web appendix 10 and 11).”

“Web appendix 11: Estimates from the main analysis repeated using neonatal mortality and post-neonatal mortality as the outcome, plotted with their confidence intervals.

Estimates for annual change (deaths per 100,000) in neonatal mortality and post-neonatal mortality 2014-2017 relative to previous trend (2000-2013) and their confidence intervals by small area deprivation quintile (most deprived=quintile 5).”